# Attending to the Mental Health of People Who Are Homeless by Mobile Telephone Follow-Up: A Systematic Review

**DOI:** 10.3390/healthcare11121666

**Published:** 2023-06-06

**Authors:** Cristina Jiménez-Lérida, Carmen Herrera-Espiñeira, Reina Granados, Adelina Martín-Salvador

**Affiliations:** 1Contrato Garantía Juvenil, University of Granada, 18071 Granada, Spain; cristina.jimenez.lerida@gmail.com; 2PhD Department of Nursing, Faculty of Health Sciences, University of Granada, 18071 Granada, Spain; cherrera_1@ugr.es (C.H.-E.); ademartin@ugr.es (A.M.-S.); 3Instituto de Investigación Biosanitaria de Granada (Ibs. Granada), 18012 Granada, Spain

**Keywords:** intervention, smartphone, mental health, people who are homeless

## Abstract

Background: More than 20% of the world’s population has no decent or suitable home. People who are homeless have more health problems than the rest of the population, especially mental health-type problems. The main objective of this study was to identify follow-up interventions by using mobile telephones to improve the mental health of people who are homeless and to analyze their efficiency. Methods: To do so, a systematic review was carried out in the Web of Science, PubMed, Scopus, Ebscohost, and PsyInfo databases. Results: Studies conclude that mobile phone use is a suitable means to improve adherence to medication and the mental health of the homeless. However, significant attempts to demonstrate health benefits by means of reliable and valid instruments that supplement qualitative satisfaction and feedback instruments appear to be lacking. Conclusions: The literature about mental health benefits through technology for people who are homeless is scarce and shows methodological limitations that can lead to failure when setting up methodologies in clinical practice.

## 1. Introduction

The European Federation of National Organisations Working with the Homeless (FEANTSA) [1] defines people who are homeless (PWHs) as those who cannot obtain or maintain a suitable and permanent home adapted to their situation due to economic reasons, social barriers, or being unable to autonomously live and are, thus, a group affected largely by social exclusion.

In today’s world, more than 20% of the population has no decent or suitable home, according to the UN [2]. In Spain, the last survey of the centers and services working with the PWHs conducted by the Spanish National Statistics Institute (INE) in 2020 demonstrated that the centers providing the PWHs with accommodation housed a mean of 17,772 people every day that year [3]. As a result of economic models that emphasize the exclusion and pressure of social models, among many other factors, the percentage of the “homelessness” phenomenon has increased in recent years [4]. 

According to Calvo et al. [5], other structural and individual risk factors exist that lead to this increase. The most significant structural risk factors include poverty, unemployment, restrictive policies to access housing, and an insufficient socio-healthcare protection system [6]. The most determining individual factors include suffering serious mistreatment, abuse, severe poverty in infancy [7], and mental health (MH) problems occurring both at the heart of families and early in adulthood [8]. 

Although a significantly large number of homeless people have a mistreatment/abuse background in infancy, there is evidence showing that a bidirectional relationship exists between trauma (including abuse) and not having a home [8,9,10,11]. Therefore, not having a home can be both the cause and consequence of trauma [12,13]. This implies extremely high rates of health problems for the people who are homeless compared to the rest of the population, especially MH problems [14,15,16]. In Spain, 16.6% of the PWHs have a serious MH problem (schizophrenia, bipolar disorder, major depressive disorder, substance abuse, serious personality disorders, and post-traumatic stress), but women have more than men [17]. Milburn et al.’s [8] remarkable findings on the relationship of trauma to mental health problems and externalizing behaviors suggest, overall, that family factors appear to be critical for understanding mental health problems and externalizing behaviors among adolescents who are homeless.

Owing to the lack of economic resources to face these MH problems, plus this population’s poor adherence and commitment to treatments [18], using technology/mobile telephones as a substitute for face-to-face treatments have been proposed in recent years [19,20,21,22] because it helps the PWHs to continue therapy and helps to lower the cost of socio-economical policies [22]. Moreover, having a mobile phone can help them keep in touch with the community, which provides mood-related benefits and facilitates the therapeutic process in this population [23]. Many studies state that the homeless who perceive more access to their social support network obtain better physical/mental health scores [24,25,26,27,28].

Hence, the main objective of this study is to identify follow-up interventions with mobile phones to improve the MH of the homeless and to analyze their efficiency. 

## 2. Materials and Methods

A systematic review was carried out of the scientific literature published about interventions for improving the MH of the homeless that employ technology (mobile phones, text messages, applications, etc.) to prevent, intervene or treat this population. The guidelines to perform systematic reviews as proposed in the PRISMA protocol (see Appendix A) were followed [29]. The study protocol was registered in PROSPERO (CRD42023422060). The level of evidence for and the degree of recommending the selected articles were evaluated following the list of the Oxford Centre for Evidence-Based Medicine [30] for quantitative studies and the JHNEBP Evidence Rating Scales [31] for qualitative ones. This was completed by two reviewers. Disagreements between these two reviewers were resolved by a third researcher.

### 2.1. Sources of Information

The search was performed in August and September 2022 in the Web of Science (WoS), PubMed, Scopus, Ebscohost, and PsyInfo databases. There were no restrictions as to the publishing date and language of articles. 

### 2.2. Inclusion/Exclusion Criteria

The selected publications met the following inclusion criteria: (a) articles in any language; (b) articles subjected to experts/peer review; (c) articles in which interventions with PWHs were made, and mobile technology was/mobile phones were used to prevent, intervene or treat MH; and (d) empirical studies/articles with original data and interventions.

The exclusion criteria were as follows: (a) editorial articles; (b) articles of systematic reviews or meta-analyses; (c) opinion articles; (d) action protocols; (e) articles of cross-sectional studies; (f) surveys; (g) nomad populations; and (h) gray literature.

### 2.3. Search Strategy

The employed search equation in databases, by limiting the search to titles, abstracts, and keywords, was as follows: 

[(smartphon* OR mobilephon* OR cellphon* OR telephon* OR *phone* OR “mobile phone technology”) AND (intervention* OR “clinical trial” OR “prevent*” OR tratment* OR adhesion* OR adherenc* OR complianc* OR colaboration* OR cooperation*)] AND [(mental* AND (health* OR hygiene*)) OR ((well*being OR welfare*) AND (emotion* OR psycholog* OR social*))] AND [homeless* OR “liv* rough” OR “emergenc* accommodation*” OR “liv* in institutions” OR “no* conventional* dwelling*” OR “lack of hous*” OR “tempor* in conventional* hous*” OR “street people” OR “people on the street*” OR “without shelter*” OR “without hom*” OR “liv* street*” OR “liv* public space*” OR “emergency supported housing” OR “night shelter*” OR “overnight shelter*” OR “no place of usual residenc*” OR “transitional support*” OR “shelter*” OR “refuge accommodation” OR “mobile hom*” OR “no* conventional* build*” OR “tempor* structure*” OR “sofa surfer*” OR “no permanent residenc*” OR “rough sleeper*” OR “roofless” OR “squatter*”]. 

### 2.4. Data Collection

The data to be extracted from the selected documents were previously defined to ensure that data would be uniformly collected. The full text of the selected articles was examined to obtain the following information: (a) title, authors, and year of publication; (b) study design; (c) objective/s; (d) participants (sample size, gender, age, origin, and type of homeless person); (e) intervention; (f) evaluation instruments; (g) intervention outcomes; (h) quality of articles; and (i) conclusions.

### 2.5. Selection Process

Having compiled all the documents, they were thoroughly reviewed to identify those articles that met the eligibility criteria. First, the title and abstract of all the selected articles were read to select them or rule them out by initial screening. Only 92 of the 435 selected articles passed the first screening. 

Second, the 92 articles were comprehensively read to, thus, complete the selection according to the aforementioned criteria. As a result, 87 articles were ruled out because they were cross-sectional studies (n = 35), MH did not intervene (n = 23), they did not resort to mobile phones in interventions, prevention, or therapy (n = 6), they were protocols (n = 6), they were systematic reviews (n = 3), they were congress summaries (n = 2), they were retrospective case reviews (n = 1), no full text was available (n = 1), or they were repeated (n = 10). The five remaining studies met all the criteria and were selected to be included in the systematic review.

Of these five articles, a reverse review was performed, and only one article was selected for our systematic review following the inclusion/exclusion criteria. Finally, six articles formed part of this systematic review. This article selection process was carried out by two researchers independently. Disagreements between these two reviewers were resolved by a third researcher.

## 3. Results

The different phases to follow this selection process appear in Figure 1.

The summary of the results of the selected studies appears in Table 1 and Table 2. 

### 3.1. Year of Publication

The six articles included in the systematic review were published from 2008 to 2021.

### 3.2. Study Design

Five of the six articles included a quasi-experimental research design, and only one had an experimental design. Of the quasi-experimental ones, two were a temporarily interrupted series, two were pre-post (one with post-evaluations at 1, 2, 3, and 4 months), and one was prospective. The six articles included in the review met methodological quality standards [30,31], and none were excluded for deficiencies in this respect (Table 1).

### 3.3. Participants

Regarding sample size, the study with the most participants was that by Fletcher et al. [45] with 191 subjects, and the smallest sample size (10 participants) was that in the study by Burda et al. [41]. All the interventions included men and women, although men predominated more than women in most (i.e., five of the six studies). Regarding their age, the studies conducted with younger subjects were those by Schueller et al. [36] and Glover et al. [32], with a mean age of 19.06 and 20.03 years, respectively. The participants in the other studies were of an older mean age, with the oldest in that of Burda et al. [41], whose mean age was 46.90 years. The participants were recruited from a wide range of settings: shelters for PWHs (3/6), churches offering assistance to PWHs (2/6), psychiatric hospitals (2/6), soup kitchens (1/6), and places on streets that PWHs frequent (1/6). All the works consider homelessness as not having a fixed nightly and suitable abode, but rather a shelter, institution, or public/private place not designed to be used as habitual accommodation to sleep in. 

### 3.4. Evaluation Instruments 

The variables measured in the studies and the evaluation instruments employed to do so considerably varied:-Four of the six studies measured the variable depression, but only two did a post-evaluation of depression. The most widely used questionnaire to do so was PHQ-9 [37], which was used in two of the four works [36,48];-Exposure to trauma and PTSD symptoms were measured in two of the six studies [32,36], but only one did a post-evaluation. The questionnaires in the two varied: the *28-Item Childhood Trauma Questionnaire* [33]; the *Traumatic Events Questionnaire* [40]; and the *20-item PTSD Checklist for Diagnostic and Statistical Manual of Mental Disorders-5*, PCL-5 [39];-One study [36] evaluated emotional regulation with a specific questionnaire: the *Difficulties in Emotion Regulation Scale,* DERS [38];-Only one study [32] measured the variable anxiety using the *PROMIS Bank V10 Anxiety measure* [34], but it was not taken into account for the post-evaluation;-Seriousness of taking different substances was measured in two studies with different questionnaires: *Addiction Severity Index-Lite,* or ASI-Lite [44], in one [41], and a Likert-type questionnaire created by the research team in the other [45];-Satisfaction with or the benefit perceived/feedback with the study was measured in all the interventions with questionnaires using the questions that the research team devised. Most were made using semistructured interviews (5/6), and only one employed a 16-item Likert-type scale questionnaire [32]. Only one of them also measured care quality with a specific questionnaire developed by the study team [48];-The variable adherence to medication was measured in two of the six interventions using different questionnaires: BARS [54] and the modified ASK-12 version [50];-Experience with technology was measured in two of the six studies, and both were with questionnaires created by research teams [36,48]. In one [48], acceptance of the technology was also measured by the *Technology Acceptance Questionnaire* [52];-Only one of the studies [48] measured perceived social support, using a specific questionnaire to do so: the *Medical Outcomes Study Social Support Survey* [51].

### 3.5. Intervention Outcomes

It can be generally concluded that mobile phone usage as a means to improve both adherence to medication and the mental health of PWHs is feasible and suitable. 

For the variables depression and PTSD symptoms, outcomes were contradictory: in the study by Schuller et al. [36], the participants underwent a few changes in clinical outcomes for depression, PTSD, and emotional regulation. However, the work by Moczygemba et al. [48] found significant differences in both depressive symptoms and barriers to adherence to medication.

Satisfaction with or perceived benefit from interventions was generally good: according to the study of Glover et al. [32], 63–68% of those surveyed stated that the intervention was beneficial. In the work by Schueller et al. [36], the participants who answered more telephone sessions gave the highest satisfaction rates. Indeed, all the participants would recommend someone else to participate, and 52% reported being very or extremely satisfied with their participation. 

The work by Fletcher et al. [45] concluded that telephone contact positively influenced both improvements in emotional problems and adherence to medications when they followed their assigned program. Similarly, in the works by Burda et al. [41] and Moczygemba et al. [48], the qualitative data revealed that unlimited access to smartphones allowed the participants to meet their social needs and remain in contact with doctors, family relations, and friends.

## 4. Discussion

This study performed a systematic review of six scientific articles that paid attention to interventions to improve the MH of PWHs by means of follow-up and mobile phone usage to summarize and combine existing knowledge about these interventions and to analyze their efficiency.

The years of publication of the six articles went from 2008 to 2021, and four of them were conducted in the last 4 years. This seems to indicate increased digitization, which reflects the society we find ourselves in and how this society also affects the homeless situation population. Notwithstanding, the number of studies that have employed such devices is still small and less recent compared to other populations, such as people with depressive disorders [55,56,57], adolescents [58,59], and people with personality disorders [60,61], among others. This, once again, seems to indicate that the homeless context population represents a forgotten gap in the research field and, consequently, in the clinical and political domain.

Obtaining large homeless situation samples is somewhat complex and is not easy to achieve owing to this population’s traveling and transitory lifestyle, and also to the clear and obvious difficulties of doing follow-ups with very high drop-out rates [18]. All this comes over in the studies included in the present review because only one sample included more than 100 participants. Nonetheless, the opportunities that new technologies provide, such as distance communication, the convenience of not having to go somewhere to be attended to, or the easy use of such technologies, can help PWHs to participate in future research to a greater extent, and have also improved adherence in the analyzed studies to obtain more replicable and encouraging results in this domain. 

The analyzed sample showed that samples formed by more men than women prevailed. In the homelessness context, this could be because women are more vulnerable than men to certain violent situations (gender violence and sexual violence, among others) and are also more exposed than the general female population to these types of violence. This is known by shelters, associations, etc., which offer resources for PWHs. As such, these women are more likely to obtain temporary accommodation or use some available resources sooner, which would benefit them and allow them to not continue in this situation [62]. However, the increase in women in homeless situations, especially young women, is relevant. This tendency has been registered in Europe by the European Observatory of the Homeless of FEANTSA [63,64], and is backed by the data obtained by the INE from 2005 to 2012. Along these lines, promoting gender equality in strategies to deal with the lack of housing should be crucial because the needs and experiences of both women and men differ. This requires personalized approaches and solutions for their problems. It would be relevant for future research to check if any differences exist in psychological mobile phone interventions in relation to participants’ gender.

The age range of the homeless included in the reviewed studies was wide, with mean ages ranging from 19 years [36] to 46 years [41]. In the different analyzed reviews that included PWHs and mental health interventions, age was not a factor that they tended to take into account [65,66]. Therefore, future studies are recommended to bear in mind the influence of age on intervention outcomes. 

Although all the reviewed studies included homeless samples, no consensus has been reached and followed up by researchers to know and determine the situation in which these people live. So using the *European Typology on Homelessness and Housing Exclusion*—ETHOS [67] as a means to improve understanding and measuring homelessness in Europe, and to provide a common “language” for transnational exchanges about homelessness, is recommended.

Regarding the evaluation instruments related to the MH variables (depression, PTSD, anxiety, etc.), the variety of tools employed in the different studies can make the comparison of their results difficult. It is worth highlighting that most of the reviewed studies did not employ standard instruments or tests to reliably evaluate these variables, but only evaluated satisfaction with the program using qualitative questionnaires designed by the research team, which is the only index to bear in mind. 

The results of this review seem to be generally encouraging. All the studies concluded that using mobile phones is a reliable and suitable means to improve adherence to medicine and the MH of PWHs. These data agree with Heaslip et al. [68], who identified connectivity via mobile phones as a very important aspect for PWHs because they provide a continuous connection with friends and family relations, which somewhat benefits MH. This is also supported by different studies that have linked social relationships with people’s improved health [69,70,71]. This improvement in MH by being connected with friends and family relations specifically comes up in the studies by Burda et al. [41], Thurman et al. [35], and Moczygemba et al. [48]. The results of the last of these works state that having unlimited access to smartphones allowed the participants to meet their social needs and to keep in contact with case managers, healthcare suppliers, friends, and family relations, and went beyond the main study proposal: that of improving healthcare coordination. 

In line with this, PWHs seem to consider that technology offers potential physical health and MH benefits because they help to keep appointments (increased adherence to medication), provide online support, and also maintain connections with friends, family relations, and health professionals. However, significant efforts to explicitly demonstrate health benefits from a clinically substantial perspective seem to be missing. This aspect has also been stressed by Heaslip et al. [68]. The work by Schueller et al. [36], which measured different variables, such as depressive symptoms, emotional regulation, or PTSD symptoms, using standard questionnaires in both pre- and post-treatments, noted a few changes in the participants’ clinical outcomes. Those authors stated that their small sample size accounted for this fact. As such, it would be worthwhile for future research lines to include reliable and valid instruments that back and supplement qualitative instruments to obtain satisfaction and feedback from programs, and to conduct studies with larger homeless context population sample sizes. 

A very interesting piece of information to bear in mind in future research works was stressed in the work by Glover et al. [32]: the homeless prefer their mobile phone functions to be totally automated and brief interventions than those requiring more participation or interaction by professionals/peers. This can guide us when preferring adherence to interventions and future research to improve the MH of the PWHs. It is quite often necessary to bear in mind what is more efficient in clinically significant terms and what is more feasible for the population with whom we are working. 

## 5. Conclusions

To conclude, we observe that the literature about MH benefits for PWHs by means of technology is scarce. It can be generally concluded that mobile phone usage to improve adherence to medication, social support, and mental health of PWHs is feasible and suitable. Due to the limitations of the studies reviewed in this paper and the lack of research on technology interventions in this population, further evidence is needed to confirm the efficacy of these interventions.

## Figures and Tables

**Figure 1 healthcare-11-01666-f001:**
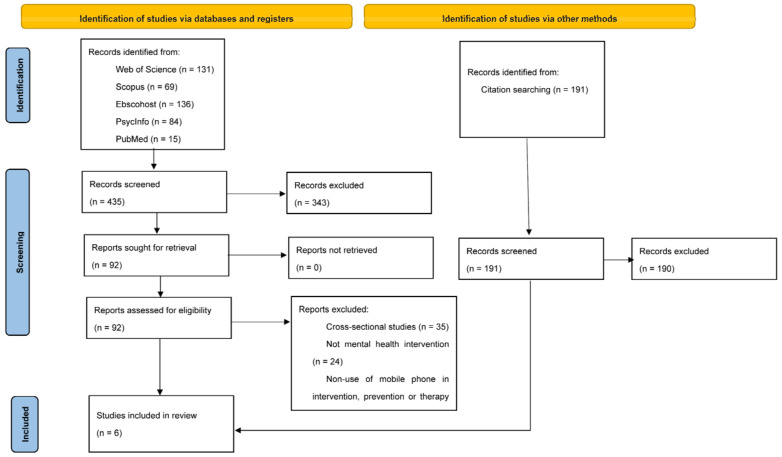
PRISMA flow chart.

**Table 1 healthcare-11-01666-t001:** Characteristics of studies and quality indices.

Authors and Year	Study Design	Objectives	Participants	Variables and Evaluation Instruments	Quality
Glover et al. (2019) [32]	Quasi-experimental	Establishing the feasibility and acceptability of providing the PWHs ^1^ with automated mental health resources by means of smartphone technology	*N* = 99 Gender: 57 men (57.58%) and 42 women (42.42%) Age: 16–25 years (*M =* 20.03 years) Origin: two PWHs shelters in Chicago (Illinois, USA) Homelessness situation: - Housing instability ^2^; - Sharing homes with other people due to loss of home or money problems; - Moving frequently; - Poor-quality home ^3^; - About to leave the temporary accommodation system.	- Physical, emotional and sexual abuse: *Childhood Trauma Questionnaire* [33]; - Anxiety by *The computer-adaptive Patient-Reported Outcomes Measurement Information System (PROMIS) Bank V10 Anxiety measure* [34]; - Depression with *The computer-adaptive PROMIS Bank V10 Depression measure* [34]; - The benefit perceived from both the study and the tools used in it was evaluated by means of the 16-item questionnaire developed by the study team (5-point Likert scale); - Participation in the study was measured by conducting daily surveys (the *Pocket Helper 2.0* application).	2b/B
Thurman et al. (2021) [35]	Qualitative	Investigating how access to smartphone technology facilitates self-management, including meeting social needs in the homeless context	*N* = 31 Gender: 22 men (70.97%), 8 women (25.80%), and 1 other (3.23%) Age: *M* = 42.7 years (*SD* = 9.67) Origin: three churches that provide the PWHs with assistance in Austin (Texas, USA) Homelessness situation: - Time as homeless: *M* = 7.4 years (range: 1 month–30 years); - 81% PWHs, sleeping on streets or in tents; - 13% in shelters; - 3% just left prison; - 3% stayed at a friend’s home.	- Semistructured interviews about participation in the study, having and using a smartphone, visits to A&E service or a hospital during the study period, and using medicines.	III/B
Schueller et al. (2019) [36]	Quasi-experimental	Evaluating the feasibility, acceptability, and preliminary benefits of a distant mental health intervention based on mobile phones with many components for young PWHs adults	*N* = 35 Gender: 23 women (65%), 11 men (31%), and 1 transgender (3%) Age: *M* = 19.6 years (*SD =* 0.85) Origin: network of PWH shelters in Chicago (Illinois, USA) Homelessness situation: - Lacking fixed regular and suitable night accommodation; - Sleeping in a PWHs shelter for at least four nights the previous week.	- Experience with mental health treatments: *Treatment Questionnaire* developed by the study team; - Experience with technology: *Technology Questionnaire* created by the study team; - Depressive symptoms: *Patient Health Questionnaire,* PHQ-9 [37]; - Emotional regulation: *Difficulties in Emotion Regulation Scale*, DERS [38]; - Present post-traumatic stress disorder (PTSD) symptoms: *PTSD Checklist for Diagnostic and Statistical Manual of Mental Disorders-5*, PCL-5 [39]; - Assessment of trauma exposure during the study period: modified version of the *Traumatic Events Questionnaire* [40]; - *Feedback Questionnaire* about the program created by the study team, which was handed out during the final sessions.	2b/B
Burda et al. (2012) [41]	Quasi-experimental	Examining the usefulness of mobile phones for collecting self-informed data as a means to monitor adherence to medicines by the homeless with psychiatric diseases	*N* = 10 Gender: 8 men (80%) and 2 women (20%) Age: *M* = 46.90 years (*SD =* 8.8) Origin: *Health Care for the Homeless* in Baltimore (Maryland, USA), patients of a psychiatric center undergoing pharmacological treatment Homelessness situation: the participants had to meet the criterion of being homeless or be at risk of becoming people in homeless situation.	- *Voxeo’s Interactive Voice Response* System to program surveys by phone daily: a 2-element survey about taking medicines and self-informed side effects conducted by the research team; - Confirmation of being diagnosed with an MH disorder: *Mini-International Neuropsychiatric Interview*, MINI [42]; - Depressive symptoms: *Center for Epidemiologic Studies Depression Scale*, CES-D [43]; - Seriousness of substance use: *Addiction Severity Index-Lite*, ASI-Lite [44]; - Feedback about the study in the final interview was created by the research team.	2b/B
Fletcher et al. (2008) [45]	Experimental	Evaluating the efficiency of three approaches for treating dual disorder patients (*people with a serious mental health disease and disorder from substance abuse*) who are homeless when recruited: integrated assertive community treatment, (IACT), assertive community treatment only (ACTO), and standard care (SC)	*N* = 191 Gender: 80% men and 20% women Age: *M* = 40 years (*SD = 9.13*) Origin: variety of settings (i.e., emergency shelters, soup kitchens, psychiatric hospitals, and places on the streets frequented by the homeless). Country not specified (US authors) Homelessness situation: - Currently in a shelter; - Living in an abandoned building; - Sleeping in a car or public place.	- Participants’ satisfaction: scale developed for this project; - Housing situation: monthly appraisal; - Psychiatric symptoms: *Brief Psychiatric Rating Scale*, BPRS [46]; - Seriousness of drinking alcohol and drug use: 3-monthly appraisal with two 5-point scales previously used in many studies [47]; - Mediators: monthly appraisal of using the service about: - Contacts with the program; - Number of days that substance abuse problems were discussed with the assigned program (substance abuse contact); - Telephone contacts with their assigned program. - If the program had helped them (dichotomic Yes/No questions): - To find permanent housing; - In activities of daily living (cooking, house cleaning); - With their emotional problems; - With adhering to medication; - With transport.	1b/A
Moczygemba et al. (2021) [48]	Quasi-experimental	Investigating the accuracy, acceptability and the preliminary results of an mHealth intervention equipped by GPS (GPS-mHealth) and designed to alert community health paramedics when the PWHs are at A&E services or a hospital	*N* = 30 Gender: 20 men (67%), 9 women (30%), and 1 other (3%) Age: *M* = 44.1 years (*SD* = 9.7) Origin: two churches that offered the PWHs assistance in Austin (Texas, USA) Homelessness situation: presently homeless situation defined as the place where someone has spent most nights in the last 30 days as follows: - 67% on the street; - 7% in a shelter; - 30% other (not specified).	- Health literacy: Brief Health Literacy Screening Tool [49]; - Depressive symptoms: PHQ-9 [37]; - Adhering to medication: Adherence Starts with Knowledge-12, ASK-12 [50]; - Social support: Medical Outcomes Study Social Support Survey [51]; - Experience with mobile technology: questions asked by the research team; - Accepting technology: modified Technology Acceptance Questionnaire [52]; - Care quality: Care Transitions Measure, CTM [53]; - Feedback about the study: semistructured interview at the end of the study.	2b/B

*Note*. ^1^ PWH = People who are homeless.^. 2^ Defined as “no fixed regular night abode, or their main night residence is a shelter, institution or public/private place not designed to be used as regular accommodation by human beings”. ^3^ For example: living in seriously overpopulated homes.

**Table 2 healthcare-11-01666-t002:** Interventions and results.

Authors and Year	Mobile Phone Monitoring Intervention	Results
Glover et al. (2019) [32]	Evaluations of perceived benefit (feedback questionnaire) after 3 and 6 months (apart from an intermediate survey 4 weeks prior to the 3-month one). Telephones had applications to promote suitable mental health and to provide recourses in real time. *Pocket Helper 2.0. (designed specifically for the study): * - Automatic daily notifications with surveys to evaluate mood and provide advice about coping and motivation; - Access to different platforms to receive emotional support: ◦ *Warm line,* a direct telephone line (offering support, tutoring, and the defense of mental health by specialists); ◦ *Koko,* a tool with access to a peers network that provides emotional support; - *Crisis Text Line,* support based on a text message at times of crisis (24 h). Support is given with text messages sent by a qualified crisis consultant; - *Pocket Helper 2.0 Support System:* brief behavioral cognitive interventions (they promote relaxation, emotional regulation, etc.). *Intellicare applications:* with 13 mini-applications. Each one centers on a singular behavioral change technique taken from behavioral cognitive therapy and positive psychology. *StreetLight Chicago*: an application with up-to-date information about social services and mental health resources for homeless youths in Chicago. The participants had to perform two daily activities: a survey of the *Pocket Helper 2.0* application and briefly mentioning the major challenge they faced the day before.	A total of 23% of the participants had problems with telephones, like theft, loss, and technological problems. Participation in the 3- and 6-month evaluations was 48% and 19%, respectively. Between 63% (30/48 at 3 months) and 68% (13/19 at 6 months) of those surveyed reported that it was a beneficial intervention. Major benefits obtained with surveys and daily advice, especially those related to motivation, overcoming difficulties, and life progress. The most used functions: ◦ Application with information about services and resources: *StreetLight Chicago;* ◦ Automated self-help system: *Pocket Helper 2.0 Support System.* The least used functions (less beneficial): ◦ Direct telephone line: *Warm Line;* ◦ Peer emotional support tool: *Koko.*
Thurman et al. (2021) [35]	The participants were given a smartphone that had a plan with text messages, calls, and unlimited data, as well as access to public transport as well. Study 1 (pilot): improve healthcare coordination and reduce its marked use by PWHs ^1^ (lasted 4 months). Study 2: improve adherence to medicines of the homeless (lasted 1 month). In the present study, final interviews were conducted with 16 PWHs who participated in Study 1 and 17 who participated in Study 2.	By having a smartphone: - The participants could more easily browse in the homelessness context and include this technology in their daily lives; - A change in the participants’ conduct, thoughts, and perceptions occurred, which empowered them, and they were capable of participating in self-management activities; - Maintained the participants’ contact with family relations and friends. In some cases, there were even reconnections after lengthy periods with no contact; - Their access to transport also facilitated social support; - Using it and knowing the time and date allowed them to establish set routines that facilitated self-management.
Schueller et al. (2019) [36]	A prepaid mobile phone with three mental health applications developed in the *Center for Behavioral Intervention Technologies*, a service and data plan, and 1 month of trainer support as three 30 min telephone sessions, plus opportunities to contact the trainer outside sessions by telephone and text messages. Trainers were qualified therapists with experience in offering treatment in homeless settings. Three telephone sessions were held: (1) orientation and identifying goals, problems, and resources; (2) control of progress and an approach for a specific theme or skill; and (3) revision of progress and discussion of the steps to follow. The skills and strategies described in the manual included the following: psycho-education, problem solving, full attention, relaxation, emotional regulation, image tests, sleep hygiene, tolerating anguish, interpersonal effectiveness, and planning security. The content of sessions was based on the principles of cognitive-behavioral approaches.	A total of 57% of the participants completed the three telephone sessions (*M* = 2.09 sessions, *SD* = 1.22). The participants sent a mean of 15.06 text messages (*SD* = 12.62) and received a mean of 19.34 text messages (*SD* = 12.70). The most popular component of the intervention was daily advice, at 64%, which indicates that they liked it considerably or a lot. Almost half the participants thought that the skills learned during a session were beneficial (48%), and almost the same number informed that they regularly used them (43%). The participants underwent a few changes in clinical outcomes: depression (*d* = 0.27), PTSD (*d* = 0.17), and emotional regulation (*d* = 0.10). Given the small sample size, none of these changes were significant.
Burda et al. (2012) [41]	The patients were given a cell phone and a free service for personal local and long-distance calls for 45 days. For 30 days, the participants received daily automatic telephone calls from the system for daily interviews. If participants could not receive the call, the system attempted to communicate with the customer by making another telephone call.	Automatic calls can act as a reminder for patients about adhering to their medication (PWHs were contacted every day and informed about taking their medication 100% of the time). Telephones helped to improve communication with their family relations and doctors. None of the 10 patients dropped out of the study or lost any mobile device. They all informed that they had taken their medication according to what they had been prescribed.
Fletcher et al. (2008) [45]	Telephone contacts at 3, 15, and 30 months. The selected participants were interviewed monthly for 30 months. They randomly received a program (IACT, ACTO, and SC).	In the three groups, telephone contact improved the efficiency of programs, especially when this contact was established during a shorter time period than the intervention. This telephone contact positively influenced the number of days they spoke about their substance abuse problems, finding stable housing, activities of daily living, improving emotional problems, adhering to medication, and using transport.
Moczygemba et al. (2021) [48]	A mobile application was used to monitor (via GPS) whether the participant attended a service at A&E or a local hospital. At that time, the researcher staff and the community paramedics team leader received notification by email; this informed community health paramedics to telephonically communicate with the participant to follow up on the visit within 2 working days and for any identified social/health needs. The paramedic also completed a report about the visit, if it could have been avoided, and what intervention could have avoided the visit to the A&E service and hospital. The intervention has two more components: (1) monthly meetings in person; (2) daily emails with reminders about adherence (if they had to take medicine that day, with “Yes” or “No” response options)	Only 19% (3/16) of reminders about visits to A&E or hospital via GPS were in line with data about the A&E service/hospital. This was mainly due to the patients not having their smartphones with them during visits, phones being switched off, or there were GPS technology gaps/problems. There was a significant difference in the depressive symptoms between the onset and at 4 months (*M* = 16.9, *SD =* 5.8 vs. *M* = 12.7, *SD* = 8.2; *p* = 0.009), and fewer barriers to taking medicines at the onset and at 4 months (*M* = 2.4, *SD =* 1.4 vs. *M* = 1.5, *SD* = 1.5; *p* = 0.003). The participants informed that the application was easy to use and emails helped them to remember to take their medicines. The qualitative data indicated that unlimited smartphone access allowed the participants to meet their social needs and to remain in contact with case managers, medical care suppliers, family relatives, and friends.

*Note*. ^1^ PWH = People who are homeless.

## Data Availability

Not applicable.

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
