# Peer review of "Attending to the Mental Health of People Who Are Homeless by Mobile Telephone Follow-Up: A Systematic Review"

_healthcare, 2023, doi:10.3390/healthcare11121666_

Round 1
Reviewer 1 Report
The researchers in this paper conducted a systematic review to explore the use of mobile phone interventions for improving mental health outcomes among homeless individuals. The study highlights the potential benefits of using mobile phone technology to enhance medication adherence and mental health outcomes, a crucial area of investigation given the high prevalence of mental health issues among the homeless population. However, the authors' review process could benefit from a more rigorous and methodical approach. Specifically, they should provide additional information on how they assessed the studies' quality and synthesize their findings more explicitly to highlight the studies' strengths and weaknesses. Overall, this paper contributes significantly to the literature, but it requires further improvements in its methodology and research synthesis to strengthen its conclusions.
Author Response
Dear Reviewer,
Authors’ response to Reviewer #1:
The researchers in this paper conducted a systematic review to explore the use of mobile phone interventions for improving mental health outcomes among homeless individuals. The study highlights the potential benefits of using mobile phone technology to enhance medication adherence and mental health outcomes, a crucial area of investigation given the high prevalence of mental health issues among the homeless population. However, the authors' review process could benefit from a more rigorous and methodical approach. Specifically, they should provide additional information on how they assessed the studies' quality and synthesize their findings more explicitly to highlight the studies' strengths and weaknesses. Overall, this paper contributes significantly to the literature, but it requires further improvements in its methodology and research synthesis to strengthen its conclusions.
Response: Thank you for your comments. We have incorporated your feedback into the article. It's marked in yellow in the Material and Method and Conclusions sections.
For more details please see the revised manuscript.
Reviewer 2 Report
Thank you for sharing your work. This was a well conducted review of a challenging area of practice. However, there are a couple of areas where I felt more depth and information was required including:
Be careful using terms like the homeless, this can promote continued stigma, try to remember it is people who are homeless
You note that the historical experience of trauma automatically implies high rates of health problems, but you do not provide any evidence for this is, please add references to justify this statement
I note that both cross sectional studies and surveys were excluded from the review but not rationale for this was provided. Please provide a rational for their exclusion
It is not clear who did the shifting – was it one author or was this independently reviewed by multiple members and then discussed? Please elaborate.
Was the protocol for the review published anywhere? If not, why... if yes, where
I was really surprised that issues affecting mobile phone usage was not explored such as owning phone, charging the phone and access to Wi-Fi – digital exclusion is a massive issue for people who are homeless and this needs to be addressed and explored more fully in the paper.
Whilst critical appraisal was undertaken there was no exploration in the results to the quality of the research in the papers included in the review apart from noting they met quality standards
The PRISMA reporting system needs to identify where in the paper the topic is covered
Author Response
Dear Reviewer,
Authors’ response to Reviewer #2:
Thank you for sharing your work. This was a well conducted review of a challenging area of practice. However, there are a couple of areas where I felt more depth and information was required including:
Response:
Thank you very much for taking the time to review our manuscript. Your comments have encouraged us to improve the manuscript and to correct several important omissions in the previous version. We respond to your comments one by one. In the revised manuscript, substantive changes have been highlighted in yellow.
Be careful using terms like the homeless, this can promote continued stigma, try to remember it is people who are homeless.
Response:
Following the Reviewer's indications, the wording has been modified in all cases.
You note that the historical experience of trauma automatically implies high rates of health problems, but you do not provide any evidence for this is, please add references to justify this statement.
Response:
We have incorporated new information in the Introduction section.
I note that both cross sectional studies and surveys were excluded from the review but not rationale for this was provided. Please provide a rational for their exclusion.
Response:
The objective of the present review was to evaluate mobile phone interventions, therefore, only clinical trials were considered and this was considered as an inclusion criterion.
It is not clear who did the shifting – was it one author or was this independently reviewed by multiple members and then discussed? Please elaborate.
Response:
The information has been included in lines 125-127.
Was the protocol for the review published anywhere? If not, why... if yes, where.
Response:
The review protocol has been registered in PROSPERO (as recommended by PRISMA). This information has been incorporated in the Material and Methods section.
I was really surprised that issues affecting mobile phone usage was not explored such as owning phone, charging the phone and access to Wi-Fi – digital exclusion is a massive issue for people who are homeless and this needs to be addressed and explored more fully in the paper.
Response:
Although this population has digital discrimination, it was not assessed in the studies included in this review. See table 3 "Variables and Evaluation Instruments".
Whilst critical appraisal was undertaken there was no exploration in the results to the quality of the research in the papers included in the review apart from noting they met quality standards.
Response:
Specific information has been incorporated in section "3.2. Study Design".
The PRISMA reporting system needs to identify where in the paper the topic is covered.
Response:
If the reviewer refers to registration in PROSPERO, the topic to which this review was assigned was "nursing". The study protocol was registered in PROSPERO (CRD42023422060).
For more details please see the revised manuscript.